

# The diagnostic utility of glycosaminoglycans (GAGs) in the early detection of cancer: a systematic review

Sarah Douglah, Reem Khalil, Reem Kanaan, Moza Almeqbaali, Nada Abdelmonem, Marc Abdelmessih, Yousr Khairalla and Natheer H. Al-Rawi

Oral & Craniofacial Health Sciences, University of Sharjah, Sharjah, United Arab Emirates

## ABSTRACT

**Background:** Glycosaminoglycans (GAGs) are a family of polysaccharides found abundantly in the extracellular matrix (ECM) of tissues. Research has indicated that the dysregulation of ECM, including changes and disruptions in GAGs, contributes to various cancer hallmarks such as metabolic reprogramming, persistent growth signals, immunosuppression, angiogenesis, tumor invasion, and metastasis.

**Objective:** This systematic review aims to evaluate the diagnostic accuracy of GAGs, including heparan sulfate (HS), chondroitin sulfate (CS), and hyaluronic acid (HA), in early detection of cancer.

**Method:** Four databases (PubMed, Scopus, EBSCO, and Ovid) were searched for studies in English within the last 15 years, involving at least 50 human participants. Using a two-stage process: identification and screening, 11 articles were selected and critically appraised using Critical Appraisal Skills Programme (CASP) checklists and Newcastle-Ottawa Scale (NOS) appropriate for each study design.

**Result:** Eleven studies met the inclusion criteria, encompassing various cancers like renal cell carcinoma (RCC), upper GI cancer (UGI), ovarian cancer, prostate cancer, breast cancer, lung cancer, colorectal cancer and oral cancer. Methodological quality was assessed using two established tools, with no studies exhibiting a high risk of bias. Heparan sulfate levels showed diagnostic potential in renal cancer with a maximum accuracy of 98.9%, achieving 94.7% specificity and 100% sensitivity. Chondroitin sulfate disaccharides emerged as a promising diagnostic marker in ovarian cancer and showed potential as diagnostic markers in renal cancer. However, there were no statistically significant differences in urinary chondroitin sulfate levels between patients and controls in prostate cancer. In breast cancer, hyaluronic acid showed moderate accuracy (AUC = 0.792) in distinguishing metastatic from non-metastatic disease, and a composite score incorporating multiple markers, including HA, showed even higher accuracy (AUC = 0.901) in detecting metastatic breast cancer. HA demonstrated moderate diagnostic accuracy for UGI cancers. Serum HA levels were significantly elevated in patients with oral cancer and pleural malignant mesothelioma and associated with tumor progression in patients with lung cancer. Elevated low molecular weight form of hyaluronan (~6 k Da HA) levels were found in colorectal cancer tissues.

**Conclusion:** GAGs hold potential as early cancer detection biomarkers. Further validation with larger, diverse populations is needed to validate their diagnostic accuracy and clinical utility.

Corresponding author
Natheer H. Al-Rawi,
nhabdulla@sharjah.ac.ae

## INTRODUCTION

Early detection of cancer is a critical factor in improving patient outcomes and reducing mortality rates (*Ahlquist, 2018*). In this context, there is an increasing interest in the identification of subclinical molecules, including polysaccharides, proteins, and receptors, that induce immune responses and indicate the early stages of cancer development (*Lepedda et al., 2022*). This correlation between cancer hallmark changes and biomarkers elucidates their potential diagnostic function and contribution to cancer progression.

Glycosaminoglycans (GAGs), a family of polysaccharides abundant in the extracellular matrix (ECM). Alterations in GAGs, as well as dysregulation of the ECM, are responsible for a variety of cancer-related characteristics, including metabolic reprogramming, sustained proliferative signaling, immunosuppression, angiogenesis, invasion, and metastasis (*Walker, Mojares & del Río Hernández, 2018*; *Deb et al., 2022*). In a previous review, *Xu, Tang & Zhang (2019)* discussed the important function of GAG molecules in identifying different types of malignancies in various sample media (*Lepedda et al., 2022*). As illustrated in *Xu, Tang & Zhang (2019)*, their involvement in numerous biological mechanisms and cell signaling led to the development of various cancer-related characteristics. Consequently, GAG is a potential diagnostic cancer biomarker for multi-cancer early detection (MCED) (*Bratulic et al., 2022*). The examination of GAGs, including heparan sulfate (HS), chondroitin sulfate (CS), keratin sulfate (KS) and hyaluronic acid (HA), has shown promise in various diagnostic approaches for a wide range of cancer types. Studies have demonstrated the predictive capacity of HS modifications in renal cell carcinoma and head and neck squamous cell carcinoma (*Gatto et al., 2016*, *2018*), while CS disaccharides have been investigated as potential biomarkers in ovarian and prostate cancers, and non-metastatic clear cell renal cell carcinoma (*Biskup et al., 2021*; *Silva et al., 2018*; *Gatto et al., 2022*). Elevated HA levels have been linked to the diagnosis and progression of breast, lung, GI, and oral cancers (*El-Mezayen et al., 2013*; *Aghcheli et al., 2012*; *Creaney et al., 2013*; *Rangel et al., 2015*; *Xing et al., 2008*). However, the diagnostic accuracy of using GAG profiles as biomarkers for early cancer detection remains unclear. This systematic review aims to address this gap by comprehensively evaluating the evidence regarding the use of GAGs in early cancer detection, assessing their diagnostic accuracy, and identifying potential biases and inconsistencies in the literature.

## METHODOLOGY

### Research question and aim of study

The research question was formulated based on the Population, Intervention, Comparison Outcome (PICO) framework: Is glycosaminoglycans (GAGs) diagnostic test accurate in the detection of early-stage cancer? This systematic review aims to assess the sensitivity, specificity, and overall diagnostic accuracy of using GAGs in detecting early-stage cancer. Moreover, to identify any potential bias or inconsistency in the use of GAGs as a diagnostic

tool for cancer detection. The protocol was registered in OSF Registries with registration DOI: https://doi.org/10.17605/OSF.IO/VCN9W.

## Search strategy

Four databases (EBSCO, PubMed, Scopus and Ovid) were used to identify publications that satisfy our eligibility criteria. Different combinations of keywords were entered in each database search engine. The search results were limited across all databases to the years 2008–2023. Additionally, we manually searched the databases for related articles that may have been missed in the electronic search. The Boolean operators utilize the following combination ("Diagnosis" AND ("Cancer" OR "Tumor") AND ("GAG" OR "Glycosaminoglycans") AND ("blood" OR "plasma" OR "serum" OR "urine" OR "saliva")

## Inclusion and exclusion criteria

The following criteria were employed to screen and assessed the search results:

### Inclusion criteria

- Sample Size: Studies with a sample size of at least 50 participants were included.
- Language: Studies that were published in English.
- Outcome measures: Studies that reported sensitivity and specificity values.
- Study design: Randomized controlled trials (RCTs), case-control studies, cohort studies, and diagnostic studies.

### Exclusion criteria

- Publication Date: Studies published before 2008.
- Studies that involved *in vitro* or animal models
- Congress and meetings publications
- Systematic reviews
- Non-English publications
- Non-GAG (proteins, receptors, genes) indicators.

## Study selection

Study selection was conducted in two stages: identification and screening.

The identification stage entailed entering the keyword combinations in the databases' search engines with the publication date limited to 15 years back (2008–2023). The results attained from each database totaled 402 articles. Additionally, we added five articles that we retrieved through a manual search. The resulting 407 records were then imported to *Endnote*, a reference management tool, where 71 articles were removed for duplication. In the screening stage, titles and abstracts were independently reviewed by two authors based on the predefined eligibility criteria. Any discrepancies were resolved by discussion with a third reviewer. Only articles that satisfied the specified criteria were included.

Full texts of potentially eligible studies were retrieved and assessed for inclusion based on study design, population, intervention, outcome and language. A total of 11 studies met

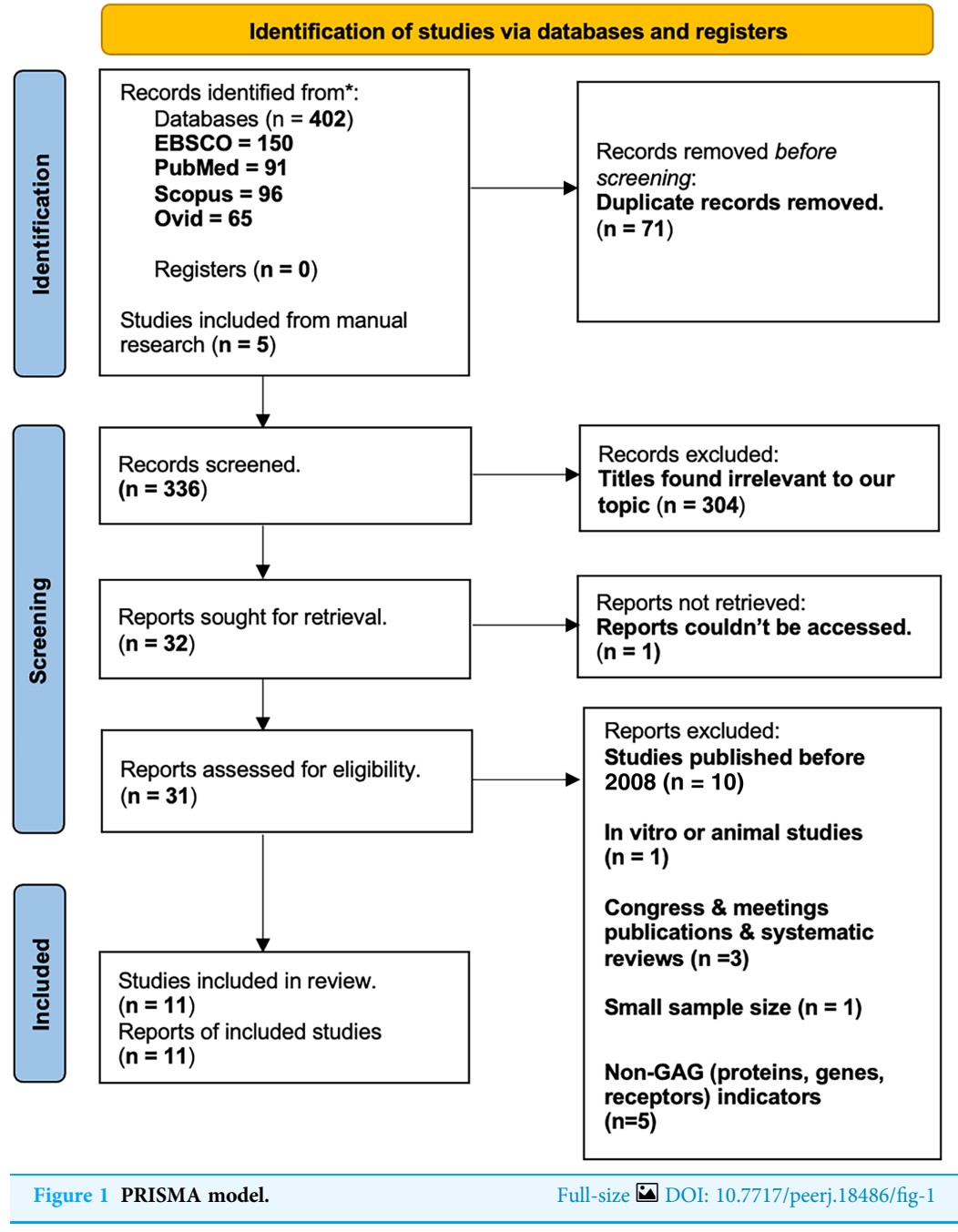

**Figure 1  PRISMA model.**               

the inclusion criteria and were included in the final qualitative synthesis (see PRISMA Flowchart–Fig. 1).

We anticipate that the publications we have selected will align with phase 2 of the Early Detection Research Network's (EDRN) five-stage biomarker development process, which includes clinical tests and validation (*Robey, 2004*). Phase 2 is concerned with the confirmation of GAG through smaller-scale clinical testing before transitioning to a larger-scale clinical study, which is delineated in phase 3. This analytical strategy ensures

**Table 1** Characteristics of the studies.

| Author/ year | Country | Study type | Population | Intervention | Outcomes | Statistical analysis | Limitations | Strength of article |
|---|---|---|---|---|---|---|---|---|
| *Gatto et al.* (2016) | Italy, Sweden | Case control | N = 50; 34 mccRCC patients + 16 controls | The accuracy and gene regulation of GAG biosynthesis to serve as non-invasive biomarker for mccRCC | Regulation of GAG biosynthesis in two independent datasets show gene expression in mccRCC vs. non-tumor-strong and significant correlations between expression fold changes in these studies and the TCGA samples | - Logistic regression -Wald z-statistics - Consensus-gene set enrichment analysis (GSA) methods: - ROC Curve: **1.** Discovery combined & plasma marker AUC-1 Specificity & Sensitivity - 100% **2.** Discovery urine marker AUC-0.966 Specificity-100% Sensitivity-84.6% | Small sample size | Good evidence (grade score 9/12 based on CASP) |
| *Gatto et al.* (2018) | USA | Case control | N = 237 218 Patients 19 healthy controls | Diagnostic & prognostic evaluation of plasma GAGs as biomarkers for RCC | Plasma GAGs are highly sensitive diagnostic and prognostic biomarkers in surgically treated RCC | -Mann-Whitney test -Bayesian estimation | Lacks generalization to the population | Good evidence (grade score 8.5/12 based on CASP) |
| *Biskup et al.* (2021) | Germany | Case control | N = 68; 28 ovarian cancer patients + 10 sepsis patients + 30 healthy volunteers | CS-based diagnosis in differentiating early-stage ovarian cancer & assess possible correlation between CS and age | - CS-bio biomarker was effective in differentiating early stage of epithelial ovarian cancer from healthy controls. - CS-bio biomarker has better sensitivity and specificity than CA125. - Presence of certain CS disaccharides was found independent of patients' ages and control. | - Shapiro-Wilk test - Mann-Whitney U-test - Binary Logistic regression - Jonckheere-Terpstra test - ROC Curve AUC (0.74–0.91) for EOC CS sensitivity of 93% & specificity of 100% | Small sample size | Good evidence (grade score 9/12 based on CASP) |

(Continued)

| | | | | | | | | |
|---|---|---|---|---|---|---|---|---|
| **Table 1** (continued) | | | | | | | | |
| **Author/ year** | **Country** | **Study type** | **Population** | **Intervention** | **Outcomes** | **Statistical analysis** | **Limitations** | **Strength of article** |
| Silva et al. (2018) | Brazil | Case control | N = 58; 44 prostate cancer patients + 14 controls | Diagnostic & prognostic evaluation of CS and HS in urine, and HA in plasma of prostate cancer patients before and after treatment. | - Serum HA was increased in cancer patients and in high-risk patients. <br> - Surgically treated patients had a significant decrease in levels of HS after surgery. <br> - No difference in urinary CS & HS between cancer patients and control | - Kolmogorov-Smirnov test <br> - Students t-test and ANOVA with Tukey's auxiliary test or Kruskal-Wallis <br> - Mann-Whitney test | Small sample size | Average evidence (grade score 6.5/12 based on CASP) |
| Gatto et al. (2022) | Sweden | Cohort | N = 127; 72 for nephrectomy for suspected RCC+ 55 for metastatic RCC therapy | Free GAG and post-surgical recurrence in non-metastatic renal cell carcinoma | Free CS-RCC recurrence ratings have the potential to significantly enhance RCC patient follow-up protocols, which could have a beneficial benefit. | - Bayesian logistic regression <br> - Bayesian additive regression Trees (BART) models <br> - ROC Curve (AUC 0.91–0.94) <br> - GAGomes presents 81% sensitivity & 80% specificity | Small number of recurrences | Good evidence (grade score 9/12 based on CASP) |
| El-Mezayen et al. (2013) | Egypt | Cohort | N = 250; 129 patients with primary breast cancer + 88 with metastatic breast cancer + 32 controls | Create a non-invasive score for the evaluation of metastasis in patients with breast cancer that is based on GAG (HA) of the ECM | Substantial increase in HA, glucuronic acid, and N-acetyl-b-D-glycosaminidase (NAG) activity in metastatic group. | - ELIZA; CA 15.3 was determined by microparticle enzyme immunoassay <br> - ROC Curve (AUC 0.792) <br> 46–86% sensitivity <br> 100% specificity | - Patients lost in cohort <br> - Follow-up period not long enough | Good evidence (grade score 10/12 based on CASP) |
| Xing et al. (2008) | China | Case Control | N = 216. <br> 84 with oral cancer <br> 65 with benign tumors <br> 67 healthy controls | Determine the levels of serum hyaluronan in patients with oral cancer and evaluate the value of serum HA. | Serum HA concentrations were significantly higher in patients with oral cancer than in patients with benign tumors and in normal controls | Serum HA concentration was measured by radioimmunoassay | Lacks generalizations to the population | Good evidence (grade score 9/12 based on CASP) |

| Author/year | Country | Study type | Population | Intervention | Outcomes | Statistical analysis | Limitations | Strength of article |
|---|---|---|---|---|---|---|---|---|
| Aghcheli et al. (2012) | Iran | Case Control | $N = 88$ 20 patients with gastric cardia cancer, 23 with gastric noncardiac cancer, 20 with esophageal squamous cell carcinoma, and 25 controls | Assessing whether hyaluronic acid and laminin may be used to identify potentially high-risk groups of upper gastrointestinal cancers | Serum hyaluronic acid and laminin concentrations in cancer cases were higher than in controls. | - Q–Q plots - Shapiro–Wilk W test. | Small sample size | Good evidence (grade score 8.5/12 based on CASP) |
| Zhang et al. (2019) | China | Cohort | $N = 322$; 184 CRC Patients, 75 Benign disease patients, 63 control patients | Assessing whether CRC-associated ~ 6 kDa HA was involved in tumor metastasis. | ~ 6 kDa HA levels in cancer tissues were positively correlated with tumor metastasis and invasion, with different invasive abilities | - Mann–Whitney rank-sum test - Student's t-test | Lacks generalization to population | Good evidence (grade score 8/12 based on CASP) |
| Creaney et al. (2013) | Australia | Cohort | $N = 164$ 96 MM patients, 26 lung cancer patients 42 patients with benign effusions | Whether pleural effusion hyaluronic acid can be used as a diagnostic and prognostic marker in pleural malignant mesothelioma | Combined biomarker panel has greater diagnostic accuracy than effusion mesothelin alone | - IBM® - SPSS® statistics - Kruskal–Wallis test | Small number of recurrences | Good evidence (grade score 9.5/12 based on CASP) |
| Rangel et al. (2015) | Brazil | Cohort | $N = 115$ 76 males; 39 females | To evaluate the association between malignant changes and HA through its expression and location in tumor tissue | Confirmation of the prognostic and diagnostic importance of HA in lung cancer | - Kolmogorov-Smirnov test - Mann-Whitney test - Kruskal–Wallis test | Patients lost in cohort | Good evidence (grade score 8/12 based on CASP) |

that the established markers, GAGs, are dependable, repeatable, and have potential in clinical trials.

## Quality assessment and data extraction

The 11 articles were analyzed and appraised using Critical Appraisal Skills Programme (CASP) checklists appropriate for each study design: cohort and case-control. Each article was given a grade score representing the strength of the article based on the questions of the CASP checklist. The articles were graded: good evidence, average evidence, or poor evidence (Table S1–*CASP Checklist*).

The data from the articles was extracted in a tabular form. The following details were included: author's names, year of publication, location of study, study type, population, intervention, outcomes, statistical analysis, limitations, and overall strength of the article (Table 1–*Evidence Table*).

The methodological quality of the included studies was assessed using validated tools appropriate for each study design. The Newcastle-Ottawa Scale (NOS) was used to evaluate the quality of the 11 observational studies: 6 case-control studies (*Gatto et al., 2016*, *2018*; *Biskup et al., 2021*; *Silva et al., 2018*; *Aghcheli et al., 2012*; *Xing et al., 2008*) and 5 cohort studies (*El-Mezayen et al., 2013*; *Gatto et al., 2022*; *Zhang et al., 2019*; *Creaney et al., 2013*; *Rangel et al., 2015*).

The quality ratings for the NOS assessment were categorized as good, fair, or poor according to the Agency for Healthcare Research and Quality (AHRQ) standards as follows:

- **Good quality:** three or four stars in the selection domain AND one or two stars in the comparability domain AND two or three stars in the outcome/exposure domain.
- **Fair quality:** two stars in the selection domain AND one or two stars in the comparability domain AND two or three stars in the outcome/exposure domain.
- **Poor quality:** 0 or one star in the selection domain OR 0 stars in the comparability domain OR 0 or one star in the outcome/exposure domain (Tables 2A and 2B).

## RESULTS

Adhering to PRISMA guidelines (Fig. 1), this systematic review initially yielded 407 studies. After applying our predefined criteria, 11 studies were included in the final qualitative synthesis. These studies encompassed a diverse range of cancer types, including renal cell carcinoma (RCC), upper gastrointestinal cancers, ovarian cancer, prostate cancer, breast cancer, lung cancer, mesothelioma, colorectal cancer, and oral cancer. Several sample types, including plasma ($n = 3$), serum ($n = 6$), urine ($n = 3$), sputum ($n = 1$), and pleural effusion ($n = 1$) were employed. The publication dates ranged from 2008 to 2023.

The methodological quality of the eleven observational studies, primarily comprising case-control ($n = 6$) and cohort designs ($n = 5$), was assessed using the Newcastle-Ottawa Scale (NOS). According to the Agency for Healthcare Research and Quality (AHRQ)

**Table 2A** Quality assessment of case-control studies and cross-sectional using the Newcastle-Ottawa Scale (NOS).

| Article | Selection | | | | Comparability | Outcome | | | Quality score |
|---|---|---|---|---|---|---|---|---|---|
| | 1 | 2 | 3 | 4 | 5 | 6 | 7 | 8 | |
| Gatto et al. (2016) | - | X | - | - | - | - | - | X | Good (6) |
| Gatto et al. (2018) | - | - | X | - | + | - | - | X | Good (7) |
| Biskup et al. (2021) | - | X | - | - | - | - | - | X | Good (6) |
| Silva et al. (2018) | - | X | - | - | - | - | - | X | Good (6) |
| Aghcheli et al. (2012) | - | - | X | X | + | - | - | X | Fair (6) |
| Xing et al. (2008) | - | - | X | X | + | - | - | X | Fair (6) |

Note:

+ : ** ; - : * ; X : 0.

**: 2 points; *: 1 point; X: 0 point

**Table 2B** Quality assessment of cohort studies using the Newcastle-Ottawa Scale (NOS).

| Article | Selection | | | | Comparability | Outcome | | | Quality score |
|---|---|---|---|---|---|---|---|---|---|
| | 1 | 2 | 3 | 4 | 5 | 6 | 7 | 8 | |
| El-Mezayen et al. (2013) | - | X | - | - | + | - | - | X | Good (7) |
| Gatto et al. (2022) | - | X | - | - | + | - | - | X | Good (7) |
| Zhang et al. (2019) | - | X | - | - | + | - | X | X | Good (6) |
| Creaney et al. (2013) | - | X | - | - | + | - | X | - | Good (7) |
| Rangel et al. (2015) | X | - | - | - | + | - | X | X | Good (6) |

Note:

+ : ** ; - : * ; X : 0.

**: 2 points; *: 1 point; X: 0 point

standards, nine studies were rated as good quality, two studies were rated as fair quality (Tables 2A and 2B). The eleven studies were also assessed using the Critical Appraisal Skills Programme (CASP) checklists. Ten studies demonstrated a low risk of bias, while

one exhibited average strength (Table 1). Notably, none of the included studies had a high risk of bias.

## Heparan sulfate

### Renal cell carcinoma

*Gatto et al. (2016)* conducted a case-control study (Italy and Sweden, $n = 50$) that examined HS and its potential for detecting metastatic RCC. Using gene expression analyses and integrated metabolic modeling, the identified genes that are involved in the biosynthesis of HS revealed evidence of a molecular connection between HS and the progression of the disease. In kidney tissues, immunohistochemistry demonstrated modifications in the HS biosynthesis proteins EXTL1 and HS6ST2 (*Gatto et al., 2016*). A combined marker of plasma and urine GAGs was able to differentiate RCC patients from controls with an AUC of 1.0. This value is considerably greater than the AUC for urine and plasma markers alone, which are 1 and 0.966, respectively. This combined marker exhibited 100% sensitivity and 100% specificity in predicting the occurrence of RCC in a validation cohort. Although urine markers demonstrated promising results (specificity: 100%, sensitivity: 84.6%), the combined and plasma markers were more significantly correlated with clinical outcomes (*Gatto et al., 2016*). In 2018, *Gatto et al. (2018)* conducted a retrospective case-control study ($n = 194$) in the United States. The study compared the glycosaminoglycan (GAG) profile of renal cell carcinoma (RCC) patients to that of healthy controls, with a particular emphasis on CS and HS using capillary electrophoresis with laser-induced fluorescence. The principal component analysis (PCA) revealed that the RCC samples were clustered in a unique manner, with 96% of the RCC samples forming separate, heterogeneous clusters from the healthy controls. The RCC samples exhibited a lower fraction of unsulfated CS (CS–0S), an increase in 4-sulfated CS (CS–4S), a higher CS charge, and a total CS concentration (*Gatto et al., 2018*). The new plasma-derived GAG score exhibited extraordinary diagnostic performance, achieving 94.7% specificity and 100% sensitivity at an optimal cutoff score of 0.87, with an AUC of 0.999 and a maximal accuracy of 98.9%. All RCC samples exhibited an elevated GAG score, regardless of tumor stage, grade, or histology. The GAG score obtained an AUC of 0.991 and a validated sensitivity of 93.5% in a validation set of 108 preoperative RCC samples. Overall survival (OS) and recurrence-free survival (RFS) were independently predicted by the total chondroitin sulfate concentration (CStot).

An optimal CStot criterion of 1 identified 14% of patients as high-risk, resulting in a lower RFS (79.1%) than low-risk patients (94.4%). A composite risk score that incorporated CStot and tumor size further refined patient stratification, identifying 22% of patients as high-risk and providing more precise recurrence predictions.

## Chondroitin sulfate

### Ovarian cancer

*Biskup et al. (2021)* in their case-control study ($n = 68$) conducted in Germany examined the potential of CS disaccharides (CS-0S, CS-2S, CS-4S, and CS-2S4S) as serum markers for early-stage primary serous epithelial ovarian cancer (EOC). The research used receiver

operating characteristic (ROC) curve analysis to evaluate the diagnostic capability of every disaccharide and the CS-biomarker. The Area Under the Curve (AUC) values reported as high (0.74–0.91) suggest that the potential for detecting EOC is promising. For the detection of EOC, the study reported encouraging results with high (AUC) values (0.74–0.91) for all four disaccharides and an AUC value of 0.98 for the CS-biomarkers. Notably, the CS-biomarker significantly outperformed the established CA-125 marker in terms of sensitivity (93% *vs.* 60%) and specificity (100% *vs.* 83%), in distinguishing cases from controls, particularly in detecting early-stage EOC. Although the results are positive, the sample size is a constraint of the study. Further validation with larger cohorts is required to confirm that CS disaccharides are diagnostically useful for EOC (*Biskup et al., 2021*).

### Prostate cancer

In 2018 *Silva et al. (2018)* conducted a case-control study ($n = 58$) in Brazil, to examine urinary CS levels in patients diagnosed with prostate cancer. Although there was no statistically significant disparity in CS levels between prostate cancer patients and control subjects, the research did detect a significant increase in urinary CS levels following hormone ablation therapy. This finding may indicate that CS needs more research to assess its expression following treatment (*Silva et al., 2018*).

### Renal cell carcinoma

As prospective biomarkers for noninvasive detection of recurrent renal cell carcinoma following surgery, CS levels in urine and blood samples were investigated in a cohort study in Sweden by *Gatto et al. (2022)*. To quantify the likelihood of recurrence, the researchers constructed three predictive models utilizing plasma, urine, and combined free GAGome samples. High area under the curve (AUC) values (0.91–0.94) were observed for the scores. To ascertain the net benefit, a decision curve analysis was performed, wherein a singular cutoff value of >30 was established. The prospective clinical utility of combined free GAGomes was indicated by the favorable sensitivity and specificity scores of 81% and 80%, respectively. However, it was acknowledged in the research that external validation and further investigations were required to establish the practicality and appropriateness of these biomarkers and recurrence scores (*Gatto et al., 2022*).

## Hyaluronic acid

### Breast cancer

In 2012, *El-Mezayen et al. (2013)* conducted a cohort study in Egypt with a sample size of 249 patients. The study found that patients diagnosed with breast cancer had significantly higher concentrations of N-acetyl-η-D-glucosaminidase (NAG) activity, CA 15.3 (a tumor marker), hyaluronic acid (HA), and glucuronic acid compared to healthy controls.
In addition, metastatic breast cancer patients exhibited even higher levels of HA (median HA = 154.5 ng/ml) than non-metastatic patients (median HA = 121.5 ng/ml). HA demonstrated the highest efficacy among the markers in distinguishing metastatic breast cancer from non-metastatic breast cancer based on ROC curves (AUC = 0.792). This suggests that HA has significant potential for the detection of metastatic disease.

The multi-biomarker composite score (MBCS) was determined to have a significantly higher AUC (0.901) than the AUCs of the individual markers (CA 15.3, HA, hyaluronidase, NAG, glucuronic acid, and glucosamine) (Multivariate Discriminant Analysis). Thus, it is possible to improve the precision of detecting metastatic breast cancer by incorporating these markers (*El-Mezayen et al., 2013*).

### Upper gastrointestinal cancers

*Aghcheli et al. (2012)* conducted a case-control study in Iran to investigate the potential of serum hyaluronic acid HA and laminin as biomarker markers for upper gastrointestinal (UGI) malignancies. The study enrolled a total of 88 participants, which included 63 individuals with UGI malignancies (gastric cardia cancer, gastric nocardia cancer, and esophageal squamous cell carcinoma) and 25 controls. Compared to controls, cancer cases were older, had a higher rate of cigarette smoking, a family history of cancer, and were more likely to be male, according to demographic analysis. The serum levels of both hyaluronic acid and laminin were significantly higher in cancer cases than in controls. Age was positively correlated with both HA and laminin levels, while cigarette consumption was associated with HA levels. However, this association was reduced after accounting for additional variables (*Aghcheli et al., 2012*). Hyaluronic acid (HA) exhibited moderate diagnostic accuracy for upper gastrointestinal (UGI) malignancies. At a cutoff value of 101 ng/ml, HA demonstrated an AUC of 0.708, a sensitivity of 80%, and a specificity of 68% in the context of esophageal squamous cell carcinoma. At a cutoff point of 62 ng/ml, laminin demonstrated moderate diagnostic accuracy for gastric cardia cancer, with an AUC of 0.828, a sensitivity of 90%, and a specificity of 60% (*Aghcheli et al., 2012*).

### Pleural malignant mesothelioma

In Australia, *Creaney et al. (2013)* conducted a cohort study that examined 96 patients diagnosed with malignant mesothelioma (MM), 26 patients with lung cancer, and 42 patients with benign pleural effusions. Effusion HA levels were considerably higher in MM patients, with an AUC of 0.89 (IQR: 0.82–0.94) for MM diagnosis, in contrast to an AUC of 0.85 (IQR: 0.78–0.90) for effusion mesothelin. The AUC of 0.92 (IQR: 0.86–0.96) of a combined model that included effusion HA and serum in addition to effusion mesothelin was significantly higher than that of effusion mesothelin alone (*Creaney et al., 2013*). In MM patients, effusion HA exhibited a biphasic distribution, with a cutoff of 75 mg/L. The median survival of patients with elevated HA levels was 18.0 months (95% CI [13.7–22.4]), which was substantially higher than the 12.6 months observed in patients with low HA levels (95% CI [8.4–16.8]; $p = 0.004$) (*Creaney et al., 2013*).

### Colorectal cancer

*Zhang et al. (2019)* conducted a cohort study in China and identified a specific low molecular weight (LMW) form of hyaluronan, approximately 6 kDa hyaluronan, in colorectal cancer tissues. In cancer tissues, elevated ~6 kDa hyaluronan levels were observed in comparison to adjacent normal mucosae ($n = 31$). These levels were substantially correlated with tumor stage, grade, and metastasis. Furthermore, the concentrations of ~6 kDa hyaluronan in serum samples from colorectal cancer patients

($n$ = 184) were significantly higher than those of healthy controls ($n$ = 63) and patients with benign diseases ($n$ = 75). The ROC curve analysis revealed an AUC of 0.931 (95% CI [0.900–0.963]) for the purpose of distinguishing CRC patients from healthy controls. The sensitivity and specificity were 95.65% and 71.43%, respectively, at the cutoff value of 19.66 ng/mL. In addition, the diagnostic accuracy of CEA and CA 19-9 was surpassed by the significant association between ~6 kDa hyaluronan levels and lymphatic metastasis, as evidenced by an AUC of 0.765 for predicting lymph node (LN) metastasis (*Zhang et al., 2019*).

### Oral cancer

To investigate the efficacy of serum hyaluronic acid (HA) as a tumor marker for oral cancer, *Xing et al. (2008)* conducted a case-control study in China. The study included a total of 216 participants, including 84 individuals with oral cancer, 65 individuals with benign lesions, and 67 healthy controls. The serum HA levels of patients with oral cancer were significantly higher (153.23 ± 73.17 ng/ml) than those of benign tumors (117.83 ± 34.56 ng/ml) and normal controls (103.98 ± 31.11 ng/ml) ($p < 0.05$). The control group and patients with benign tumors did not exhibit any significant differences ($p > 0.05$). In contrast to patients with early stages (I and II), those with advanced stages of oral cancer (III and IV) exhibited substantially elevated HA levels ($p < 0.05$). Stages I and II and stages III and IV did not exhibit any significant differences in HA values ($p > 0.05$).Serum HA levels decreased in a subset of patients ($n$ = 43) as a result of treatment; however, this difference was not statistically significant ($p > 0.05$) (*Xing et al., 2008*).

### Lung cancer

Out of the 115 total participants, *Rangel et al. (2015)* evaluated 90 sputum samples and 46 lung cancer (LC) tissue samples from confirmed LC patients. In comparison to normal lung tissue, tissue hyaluronan (HA) expression was significantly greater in LC tissue, particularly in squamous cell carcinoma (SCC) ($p < 0.001$). SCC exhibited the highest HA levels, followed by adenocarcinoma (AD) and large cell carcinoma (LCC) ($p = 0.01$). In the N1 and T4 stages, elevated HA expression was significantly associated with tumor progression (R = 0.31; $p = 0.05$) and increased microvessel density (MVD) (R = 0.6; $p = 0.02$). Reduced disease-free survival and overall survival (OS) were associated with elevated HA levels ($p = 0.02$). The median survival of patients with HA concentrations below 692.1 mg/mg was 72 months, while that of patients with higher HA concentrations was 52 months ($p = 0.02$). The HA levels in sputum samples ($n$ = 90) were significantly elevated in LC patients ($p < 0.001$), with the maximum levels observed in SCC ($p = 0.01$). The ROC analysis demonstrated that sputum HA levels could differentiate LC patients from healthy controls with an AUC of 0.821, a sensitivity of 66%, and a specificity of 80%. There was a significant relationship between the levels of HA in sputum and cancer tissue (R = 0.5; $p = 0.02$) (*Rangel et al., 2015*). The GAG biomarkers that were examined in various malignancies were summarized in Table 3.

**Table 3 Summary: Glycosaminoglycans biomarkers in cancer.**

| Cancer type | GAG biomarker(s) | Samples | Potential clinical application | Key findings |
|---|---|---|---|---|
| Metastatic clear cell renal cell carcinoma (mccRCC) | HS, CS | Plasma and urine ($n = 50$) | Diagnostic marker, especially in combination | Increased HS expression is a potential diagnostic marker, and combining urine and serum analysis improves accuracy (*Gatto et al., 2016*). |
| Renal cell carcinoma (RCC) | HS, CS | Plasma ($n = 194$) | Diagnostic and prognostic markers | The plasma-derived GAG score, with an AUC of 0.999, demonstrated high diagnostic accuracy for RCC. The CStot was identified as an independent prognostic marker (*Gatto et al., 2018*). |
| Ovarian cancer | CS disaccharides | Serum ($n = 68$) | Early-stage diagnostic marker | CS disaccharide levels outperform the CA-125 marker in identifying early-stage ovarian cancer (*Biskup et al., 2021*). |
| Prostate cancer | CS | Urine ($n = 58$) | Diagnostic and prognostic marker, post-screening monitoring | No significant difference in initial CS levels between cancer and controls, but CS levels increased after hormone ablation therapy (*Silva et al., 2018*). |
| Clear renal cell carcinoma | CS | Urine, plasma ($n = 127$) | Diagnostic marker for recurrence | Combined analysis of free GAGs in urine and plasma showed promising predictive potential for recurrence (*Gatto et al., 2022*). |
| Breast cancer | HA, NAG, Glucuronic Acid | Serum (non-metastatic: 129, metastatic: 88) | Diagnostic marker, combined analysis | HA showed potential for metastasis detection. Combining HA with other markers (CA 15.3, NAG, *etc.*,) significantly improved diagnostic accuracy (*El-Mezayen et al., 2013*). |
| Upper gastrointestinal cancers | HA, Laminin | Serum ($n = 88$) | Diagnostic marker | Both HA and laminin demonstrated moderate diagnostic accuracy for UGI cancers (*Aghcheli et al., 2012*). |
| Pleural malignant mesothelioma (MM) | HA, Mesothelin | Serum, effusion ($n = 96$) | Diagnostic and prognostic marker | Effusion HA levels were significantly higher in MM, achieving an AUC of 0.89 for diagnosis. A combined model with serum mesothelin increased the AUC to 0.92 (*Creaney et al., 2013*). |
| Colorectal cancer | ~6 kDa Hyaluronan | Serum ($n = 322$) | Diagnostic marker | Elevated ~6 kDa hyaluronan levels in cancer tissues and serum associated with tumor progression, diagnostic potential (AUC = 0.931), predictive value for LN metastasis (AUC = 0.765) (*Zhang et al., 2019*). |
| Oral cancer | HA | Serum ($n = 216$) | Diagnostic marker | Elevated HA serum levels in oral cancer patients compared to controls. HA levels were higher in patients with advanced stages of oral cancer (*Xing et al., 2008*). |
| Lung cancer | HA | Sputum ($n = 90$), tissue ($n = 46$) | Diagnostic marker | Sputum and tissue HA expression was significantly higher in lung cancer patients, especially those with SqCC (*Rangel et al., 2015*). |

# DISCUSSION

Glycosaminoglycans, linear polysaccharides, are essential in extracellular matrix, supporting structural support and facilitating cell signaling, with main types including HS, CS, hyaluronic acid (HA), and keratin sulfate (KS) (*Miyamoto et al., 2011*).

## Heparan sulfate

HS, a complex glycosaminoglycan, is essential for the progression of malignancy. *Gatto et al. (2016)* investigated the diagnostic potential of the overall glycosaminoglycan (CS/HS)

ratio in metastatic clear renal cell carcinoma (RCC). The role of GAGs in the regulation of biosynthesis and the remodeling of the extracellular matrix, both of which contribute to tumorigenesis, was underscored by this approach, which was supported by a large sample size and independent validation. The study did, however, recognize its limitations, which include the potential for confounding factors like renal inflammation and the possibility that altered expression of individual GAGs could also be a characteristic of malignancy. The study's findings regarding the diagnostic potential of GAGs were substantiated by analyses of clinical outcomes in validation cohorts, despite these limitations, indicating alignment with plasma marker and combined marker scores (*Gatto et al., 2016*). This implies that, although the aggregate GAG ratio may have diagnostic value, additional research is required to completely understand the contribution of individual GAG components, including HS and its modifications, to tumor progression and their potential as biomarkers for early detection and monitoring. *Gatto et al. (2018)* conducted a subsequent study that expanded upon the diagnostic potential of GAGs in renal cell carcinoma (RCC), in addition to the findings from *Gatto et al. (2016)*. The GAG score derived from plasma, which has a high specificity (94.7%) and sensitivity (93.5%) in distinguishing RCC from healthy controls. The biomarker's potential for RCC diagnosis is evident in the high AUC values (0.999 in the discovery cohort and 0.991 in the validation cohort). It is crucial to note that the GAG score was not influenced by the tumor burden, which implies that plasma GAG alterations may be influenced by broader biological mechanisms, such as immune or stromal responses. Furthermore, the inclusion of total chondroitin sulfate (CS tot) as an independent prognostic factor for both overall survival (OS) and recurrence-free survival (RSF) demonstrated. The C Stot-based risk stratification enabled more precise predictions, indicating that plasma GAG components such as CS tot could provide substantial clinical value in patient management. This minimally invasive biomarker improves risk assessment and prognosis.

Overall, these results suggest that the plasma GAG score, particularly when combined with CS tot, has substantial potential as a diagnostic and prognostic tool in RCC. Nevertheless, the retrospective design and the variability resulting from the use of different sampling methodologies are among the limitations, underscoring the necessity of standardized assays and comprehensive assessments of baseline plasma GAG levels in a variety of populations. Despite the fact that additional prospective studies are necessary to determine their therapeutic relevance, the findings underscore the significant potential of plasma GAGs as significant diagnostic and prognostic instruments in RCC (*Gatto et al., 2018*).

## Chondroitin sulfate

Chondroitin sulfate (CS) disaccharide profiles are emerging as potential diagnostic and prognostic biomarkers for various cancers. In ovarian cancer, a novel serum marker called "CS-bio," based on the relative abundance of specific CS disaccharides, shows promise for early-stage detection (*Gatto et al., 2018*). This finding is particularly significant due to the limitations of the established marker CA-125 in early-stage disease. The observed increase in CS-4S and decrease in CS-0S disaccharides in ovarian cancer patients, likely due to

elevated CHST11 activity, corresponds with prior research demonstrating increased CS disulfation in ovarian cancer tissues (*Oliveira-Ferrer et al., 2015*; *Vallen et al., 2012*). The promising results of CS-bio in ovarian cancer highlight the potential of CS disaccharide analysis in cancer detection. Nevertheless, its applicability might be specific to certain cancer types. In prostate cancer, for instance, overall CS levels did not differ significantly between patients and controls (*Silva et al., 2018*), suggesting a more nuanced role for CS in different cancers. However, a prospective study on renal cell carcinoma (RCC) demonstrated changes in plasma and urine free GAGomes, including CS disaccharides, associated with tumor recurrence after surgery (*Gatto et al., 2016*). The high predictive value of this multi-analyte approach suggests that analyzing CS disaccharide profiles from various sources could be valuable for developing robust cancer biomarkers across different malignancies.

## Hyaluronic acid

Hyaluronic acid (HA) is a glycosaminoglycan with a multifaceted role in cancer progression. A study on breast cancer explored a multi-biomarker composite score (MBCS) incorporating HA, CA 15.3, hyaluronidase, NAG, glucuronic acid, and glucosamine (*El-Mezayen et al., 2013*). This MBCS showed significantly improved accuracy for detecting metastatic breast cancer, with an AUC value of 0.901 compared to the already established individual marker CA 15.3 with an AUC of 0.694 (*Cheung, Graves & Robertson, 2000*) suggesting a potential for enhanced diagnostic power through a combined approach. Among these markers, HA appeared as a potential marker for diagnosing metastasis, with a high AUC value of 0.792. HA also plays a crucial role in breast cancer progression. Increased HA production by tumor cells and the stroma creates a supportive microenvironment, facilitating tumor cell migration and hindering immune cell infiltration through an HA pericellular coat (*Necas et al., 2008*; *Yabushita et al., 2004*). Degradation of this coat by hyaluronidase can enhance immune attack but also generate fragments that promote angiogenesis (*Ravi et al., 2000*). A low molecular weight form of hyaluronan (~6 kDa HA) was identified in colorectal cancer (CRC) tissues by *Zhang et al. (2019)*. Concentrations of ~6 kDa HA were found to be significantly elevated in cancerous tissues when compared to adjacent normal mucosa, suggesting its possible involvement in development and progression of CRC. Additionally, serum levels of ~6 kDa HA were significantly higher in CRC patients than in healthy controls, suggesting its potential utility as a non-invasive biomarker for early detection and monitoring of the disease. The increased concentrations of ~6 kDa hyaluronan in cancer tissues and its correlation with tumor stage, grade, and metastasis highlight its connection with disease progression. Moreover, the identification of ~6 kDa hyaluronan in serum samples suggests its potential role as a non-invasive biomarker for early detection and monitoring of colorectal cancer. The diagnostic accuracy of ~6 kDa hyaluronan, indicated by the high AUC value of 0.931, exceeds that of conventional markers like CEA (0.707) and CA 19-9 (0.677), as confirmed in previous metanalyses (*Liu et al., 2014*; *Fernandez-Fernandez, Tejero & Tieso, 1995*). The sensitivity and specificity of ~6 kDa hyaluronan, recorded at 95.65% and 71.43%, respectively, highlight its potential as a significant diagnostic tool. Furthermore, the

association of ~6 kDa hyaluronan with lymphatic metastasis underscores its promise in predicting disease progression and guiding treatment decisions. *Aghcheli et al. (2012)* found that serum hyaluronic acid (HA) and laminin have moderate diagnostic accuracy for upper gastrointestinal (UGI) cancers. HA had a sensitivity of 80% and an AUC of 0.708 for esophageal squamous cell carcinoma, while laminin showed an AUC of 0.828 and a sensitivity of 90% for gastric cardia cancer. These findings suggest that neither biomarker is considerably raised, although they could still aid in distinguishing UGI cancer cases from controls (*Aghcheli et al., 2012*). Hyaluronic acid displayed a fairly good diagnostic accuracy for esophageal squamous cell carcinoma and gastric noncardiac cancer, whereas laminin showed comparatively lower effectiveness. However, laminin showed stronger utility for gastric cardia cancer, suggesting it may be a subtype-specific biomarker. HA and laminin are not exclusive to UGI cancers and can be elevated in non-malignant conditions such as liver disease and inflammation, but their diagnostic performance suggests they may be used in detecting high-risk patients. The study conducted by *Creaney et al. (2013)* confirms the diagnostic significance of effusion hyaluronic acid (HA) and mesothelin for malignant mesothelioma (MM). Soluble mesothelin, a well-characterized biomarker for MM identified in 2003 (*Robinson et al., 2003*). However, in recent meta-analyses, mesothelin alone has too low for early diagnosis, suggesting the need for further biomarker research (*Hollevoet et al., 2012*). The findings indicated that effusion HA and mesothelin levels were considerably higher in MM patients, which is consistent with earlier findings (*Roboz et al., 1985*; *Pettersson et al., 1988*; *Grigoriu et al., 2009*). The study found that a combination of effusion HA and mesothelin, both in the effusion and serum, yielded superior diagnostic accuracy compared to mesothelin alone, achieving an AUC of 0.92. This indicates that the integration of multiple biomarkers can improve the detection of MM, thereby overcoming a significant limitation associated with the use of individual markers. Serum HA proved to be non-diagnostic; however, effusion HA displayed a biphasic distribution in MM patients, with levels surpassing 75 mg/L associated with improved survival outcomes. This contrasts with various cancers such as prostate, gastric and breast cancers, where increased HA typically signifies a poor prognosis (*Ghatak et al., 2010*; *Jang et al., 2011*; *Montgomery et al., 2012*). In MM, elevated effusion hyaluronic acid levels were associated with a median survival of 18 months, in contrast to 12.6 months for individuals with reduced hyaluronic acid levels ($p = 0.004$). The reasons for this phenomenon remain unclear; however, it may relate to the distinct mechanisms by which MM tumors synthesize HA in comparison to other cancers, indicating that HA could play a protective role by inhibiting tumor dissemination within the pleura. The results of this study have important clinical implications for the management of MM patients. The combined use of effusion HA and mesothelin could improve the accuracy of diagnosis, leading to earlier detection and more effective treatment. The study conducted by *Xing et al.'s (2008)* demonstrates that serum HA levels were significantly elevated in oral cancer patients compared to individuals with benign tumors and controls, suggesting its potential as a tumor biomarker. However, the absence of specificity, due to increased HA in non-malignant conditions such as liver disease and rheumatoid arthritis, limits its diagnostic utility (*Engström-Laurent et al., 1985*; *Engström-Laurent & Hällgren, 1985*). The study found a correlation between serum HA

levels and disease stage, with higher HA concentrations in stages III and IV compared to stages I and II. This suggests that HA may reflect tumor progression and metastatic status, as seen in other malignancies (*Toole, 2004*). However, HA did not exhibit significant differences between earlier and advanced stages of oral cancer (stages I & II *vs*. III & IV), and there was considerable heterogeneity in HA values among patients. This indicates that although HA may increase with tumor burden, it is not a reliable marker for early diagnosis or as an accurate method for tumor staging. Changes in HA levels did not show a significant correlation with treatment response, suggesting limited prognostic value in assessing therapeutic results. While serum HA may serve as a supplementary diagnostic tool for oral cancer, its effectiveness in staging and monitoring treatment outcomes remains unclear and requires additional research. A study by *Rangel et al. (2015)* on lung cancer demonstrated that hyaluronan (HA) levels are significantly elevated in lung cancer tissues, particularly in squamous cell carcinoma (SCC), where HA expression correlates with advanced tumor stages and poor survival outcomes. Notably, HA concentration serves as a prognostic marker, with levels above 692.1 mg/mg associated with reduced median survival. In addition to tissue analysis, they found that HA levels in sputum from lung cancer patients were significantly higher than in cancer-free individuals and healthy controls. This study established HA's diagnostic potential, as ROC analysis indicated an area under the curve (AUC) of 0.821, providing 66% sensitivity and 80% specificity for distinguishing lung cancer from healthy individuals. These findings suggest that sputum HA levels can effectively identify patients with lung cancer, highlighting its potential as a non-invasive biomarker (*Rangel et al., 2015*). However, challenges remain in translating these findings to clinical practice. The generalizability of some studies is limited due to small sample sizes or lack of information on control groups. Additionally, while the multi-biomarker approach shows promise, its clinical utility and cost-effectiveness need further validation. Larger-scale studies and rigorous validation are necessary to confirm the clinical utility of HA as a diagnostic and prognostic biomarker, paving the way for its potential as a therapeutic target in various cancers.

## CONCLUSION

This systematic review underscores the diverse roles of GAGs in cancer biology and their emerging potential as diagnostic and prognostic biomarkers. The studies evaluated provide compelling evidence for the association of specific GAG profiles with cancer progression, metastasis, and patient outcomes across various malignancies, including renal, ovarian, breast, lung, GI, and colorectal cancers. The identification of novel serum markers like CS-bio in ovarian cancer and the diagnostic significance of distinct CS sulfation patterns in breast cancer further emphasize the clinical relevance of GAGs in oncology.

### Limitation of this study

The heterogeneity of GAG structures and lack of standardized analytical techniques hinder the accurate identification and quantification of cancer-specific GAG profiles. Developing standardized methods for GAG extraction, purification, and analysis is essential for

improving the reproducibility and reliability of biomarker studies. Additionally, the relatively small sample sizes and varying methodologies of the included studies highlight the need for larger-scale, prospective studies to validate the diagnostic and prognostic utility of GAGs in diverse patient populations. Moreover, our selected articles are primarily focused on certain areas of the world, therefore rendering the current results evaluated in this review not fully representative of the global population. This is a limitation that needs to be addressed, as biomarker performance may vary across different demographics due to genetic, environmental, and lifestyle factors. Future studies should aim to include larger, and more diverse populations to ensure the generalizability of the findings. Despite these challenges, the potential of GAGs as biomarkers for early cancer detection and disease monitoring remains significant. Further research should focus on developing advanced analytical techniques, potentially incorporating machine learning and artificial intelligence, to refine GAG profiling and improve diagnostic accuracy. A deeper understanding of the mechanisms underlying GAG dysregulation in cancer is crucial for identifying novel therapeutic targets and developing personalized treatment strategies.

### Funding

The authors received no funding for this work.

### Competing Interests

The authors declare that they have no competing interests.

### Author Contributions

- Sarah Douglah conceived and designed the experiments, performed the experiments, prepared figures and/or tables, and approved the final draft.
- Reem Khalil conceived and designed the experiments, analyzed the data, prepared figures and/or tables, authored or reviewed drafts of the article, and approved the final draft.
- Reem Kanaan performed the experiments, authored or reviewed drafts of the article, and approved the final draft.
- Moza Almeqbaali performed the experiments, prepared figures and/or tables, and approved the final draft.
- Nada Abdelmonem analyzed the data, prepared figures and/or tables, and approved the final draft.
- Marc Abdelmessih conceived and designed the experiments, authored or reviewed drafts of the article, and approved the final draft.
- Yousr Khairalla conceived and designed the experiments, analyzed the data, prepared figures and/or tables, and approved the final draft.
- Natheer H. Al-Rawi conceived and designed the experiments, analyzed the data, authored or reviewed drafts of the article, and approved the final draft.

### Data Availability

This is a systematic review/meta-analysis.

## Supplemental Information

Supplemental information for this article can be found online at http://dx.doi.org/10.7717/peerj.18486#supplemental-information.

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
