# Peer review of "The diagnostic utility of glycosaminoglycans (GAGs) in the early detection of cancer: a systematic review"

_PeerJ, doi:10.7717/peerj.18486_

## Round 0.1 · original submission · Minor Revisions

Dear authors,

Thanks for submitting this interesting work that addresses a topic deserving more attention. I warmly recommend you follow the suggestions of the reviewers to improve your manuscript.

Reviewer 1 ·

Basic reporting

1. The authors identified a clear evidence gap and the manuscript is overall qell written and clear. However, they fail to cite prior review work on this, see https://link.springer.com/chapter/10.1007/978-3-030-99708-3_6 and https://pubmed.ncbi.nlm.nih.gov/30905465/

2. I suggest to emphasize that the review is intended for noninvasive diagnostics. This could be done already within the title (e.g. “Evaluating the evidence regarding the use of GAGs in the noninvasive early cancer detection of cancer: a systematic review”) and in Introduction (e.g. line 55: “However, the diagnostic accuracy of using GAG profiles, or GAGomes, as noninvasive biomarkers for early cancer detection remains unclear” and line 57 “evaluating the evidence regarding the use of GAGs in noninvasive early cancer detection”

3. I suggest to reduce the Discussion focusing how the evidence gap identified by the authors is reduced by this review rather than the biological mechanisms.

Experimental design

1. I have strong concerns regarding the choice of the query, which affects the claim that the present study qualifies as a systematic review. The query is significantly different between the four databases, with a focus for oral cancers and saliva that the authors should justify.

Considering that the authors aim is “evaluating the evidence regarding the use of GAGs in early cancer detection” I would expect:
a. One term relating to cancer
b. One term relating to early detection or dx
c. One term relating to GAGs
d. One term relating to noninvasiveness (although not explicitly mentioned in the authors’ aim, it is implicit in the introduction and relates to my previous point in Basic Reporting)

Instead, taking PubMed as an example, the authors propose this query:

PubMed keywords:
(Diagnostic accuracy" OR "Cancer detection" OR Tumor Microenvironmentî) AND
(glycosaminoglycans" OR "GAGomes" OR "Liquid Biopsies" OR Biomarkers)
AND
(Extracellular Matrix")
AND
(Non-invasive")

The query proposed by the author appears very strict and it does not seem to capture the breadth of studies on GAGs for noninvasive cancer dx. The proposed query returned only 29 results when I tested it in Pubmed. Since the first term includes “OR” for cancer detection, many results relate to liver disease. By enforcing the third term “Extracellular matrix”, which I would regard as unnecessary (unless the authors intended to group it with “Liquid biopsies” or “biomarkers”, in which case the query is constructed incorrectly), the query excludes 1015 (!) studies. The fourth term also limits results significantly. By simply replacing it with “blood” OR “plasma” OR “serum” OR “urine” OR “saliva”, the original query jumps to 156 results according to my test.

Under these premises, I am worried that the authors missed out on several relevant studies, making the review non-systematic. Using the last query, I identified 7 studies that appear to qualify for the review, for example:

o https://onlinelibrary.wiley.com/doi/full/10.1111/j.1442-2042.2012.03086.x
o https://pubmed.ncbi.nlm.nih.gov/22153533
o https://www.ncbi.nlm.nih.gov/pmc/articles/PMC7385127/
o https://pubmed.ncbi.nlm.nih.gov/36848607/
o https://pubmed.ncbi.nlm.nih.gov/35911082/
o https://febs.onlinelibrary.wiley.com/doi/full/10.1111/febs.14859
o https://pubmed.ncbi.nlm.nih.gov/28719902/

This is a pity because I agree with the authors that there is a significant evidence gap in this field, and the rest of the methodology adopted by the authors is sound and worthy of publication. Therefore, I would strongly encourage the authors to repeat the systematic review with a consistent query constructed to accurately capture the aim of authors with the present study.

I am aware that changing the query at this stage is a post-hoc modification compared to the protocol deposit in OSF (anyway OSF registration appears quite recent, so it might be post-hoc to begin with – the authors should clarify if the protocol was deposited before or after the review was performed on line 69). But I would rather specify the post-hoc modification of the query (and PRISMA admits such instances as long as the authors are transparent about it), then risking compromising the systematic nature of the review.

2. While I share that the research question in line 65 is valid, speaking of diagnostic accuracy for a test outside a well defined target population and target condition is not useful for any clinical setting. For example, it is very different to use GAGs to find cancer early in primary vs. secondary prevention, i.e. among completely healthy asymptomatic people vs. among patients previously curatively treated for cancer and now presumed healthy vs. patients with active disease at risk of progession. The accuracy metrics of GAGs depend on the patient population. The authors make a good effort to distinguish between case-control (low level of evidence) vs. cohort (higher level of evidence) studies – I would ask the authors to clarify what they mean by “diagnostic study”. But they should emphasize that “diagnosis” in this review appear to encompass many and quite diverse clinical settings: screening (asymptomatic healthy), diagnostic work-up (symptomatic), surveillance (post-curative treatment), and monitoring (active disease under treatment) [see: https://www.degruyter.com/document/doi/10.1515/dmpt-2023-0056/html ]. The authors should make explicit their assumption that they consider suitable controls healthy individuals even in settings where these are not the actual controls, e.g. using GAGs to detect early stage bladder cancer vs. not in a patient with hematuria is only approximated by a study in which GAGs are measured in early stage bladder cancer vs. healthy. The accuracy will contain bias because healthy individuals have “more normal” GAGs than individuals without bladder cancer but hematuria because it is reasonable to assume that whatever caused hematuria might affect GAGs, leading to for example an underestimated specificity.

While, these assumptions for controls and cases are not per se wrong, the authors should be made explicit in the Methodology. In addition, I encourage the authors to refer to https://edrn.nci.nih.gov/about-edrn/five-phase-approach-and-prospective-specimen-collection-retrospective-blinded-evaluation-study-design/ to define the level of evidence of the included studies, since it appears to me that with the exception of one Phase 3 study, all other studies are still in Phase 1 (Discovery).

3. The authors report AUC, which I agree being the correct statistics since the cut-off is a much later decision based on clinical utility usually determined in Phase 4 or even Phase 5 studies, rendering the meaning of sensitivity and specificity rather moot. I would encourage the author to revise the outcome measure to AUC and to report this metric consistently (together with sample size) in their result reporting.

4. The authors should specify the exact period for exclusion (15 years ago from which date?)

Validity of the findings

1. PRISMA dictates that borderline included studies should be listed and discussed (item 16b). I could not see these in the lines indicated by the authors in their checklist.

2. I am concerned that the authors report results related to studies outside their stated aim. Specifically, they include results that relate to prognosis and not diagnosis (e.g., from the Abstract, “Elevated plasma HA levels were associated with poor survival in metastatic cases.”) or to enzymes and not GAGs (e.g. “Hyaluronan synthase 2 expression “. I would strongly recommend the authors to exclude prognostic studies or non-GAG related results from the review, since it is outside the stated aim. I do not see how Yang et al., Purushothaman et al., and Wang et al. studies are eligible for the review since they do not measure GAGs, as well as Svensson et al. that do not measure GAGs noninvasively.

3. In line 174, Gatto et al 2016 examined 77 not 50 patients.
4. In line 210, Biskup et al 2021 examined 58 not 68 patients.
5. In line 226, Silva et al. did find a statistical difference in serum HA, which is not reported.
6. The multicancer study by Bratulic et al. is included but not discussed, which is rather odd considering that it is the only one achieving Phase 3 LoE and it is by far the largest of all studies included in the review.
7. In Table S1, Yang et al. should have a score of 7/12 not 7/9

Reviewer 2 ·

Basic reporting

The systematic review delves into the diagnostic accuracy of Glycosaminoglycans (GAGs), focusing on their potential as early biomarkers for cancer detection. The review is comprehensive and well-structured, offering valuable insights into the role of GAGs, such as heparan sulfate (HS), chondroitin sulfate (CS), and hyaluronic acid (HA), in various cancers.

Experimental design

The study's methodology is robust. It incorporates a well-defined search strategy across four major databases (PubMed, Scopus, Wiley, and Ovid) and uses established critical appraisal tools (CASP and NOS) to assess the quality of the included studies. This rigorous approach ensures that the findings are based on high-quality evidence.

The review covers a range of cancers, including breast, renal, head and neck, ovarian, myeloma, bladder, prostate, and oral cancers. This diversity enhances the generalizability of the findings and highlights the broad potential of GAGs as diagnostic markers.

Validity of the findings

The results are clearly presented, with a focus on the diagnostic potential of each GAG in specific cancer types. For instance, the association of heparan sulfate with head and neck cancer and renal cancer is well-articulated, as is the potential of chondroitin sulfate disaccharides in ovarian and renal cancers. The discussion of hyaluronic acid’s role in breast cancer and oral cancer progression further underscores the importance of these biomarkers.

Additional comments

Limitations:
Limited Study Pool: The review includes only 11 studies, which might limit the comprehensiveness of the findings. Although the selected studies are of high quality, the small number might restrict the strength of the conclusions drawn.

Population Diversity: While the review acknowledges the need for validation in larger and more diverse populations, the current findings are based on studies that may not fully represent the global population. This limitation is crucial, as biomarker performance can vary across different demographics.
This needs to be discussed.

---

## Round 0.2 · Minor Revisions

Thanks for working on the manuscript according to reviewers' suggestions.
Please consider the last suggestions from reviewer 1.

Reviewer 1 ·

Basic reporting

I am satisfied with the response with the exception of the following comment:

Comment 2 – Thank you for your recommendation. As it seems reasonable to assess GAG molecules non-invasively, our search criteria yielded us with 2 strong studies that test for GAG using tissue samples. Rangel et al (http://dx.doi.org/10.1590/1414-431X20144300
) used tissue samples and sputum, while Miyamoto et al ( 10.1136/jclinpath-2011-200231) only used tissue samples. Therefore, we can’t classify the intervention in the review as non-invasive

The search criteria used in this study are related to biofluids. If the results of the review include papers outside the search criteria, then the rationale for this deviation from the search criteria should be clearly stated in the Results. This choice however calls into question if this is a systematic review because "tissue" was not included in the search criteria. I am sure that the authors appreciate that the strength of this paper is its systematic review, and by definition when the search criteria do not include tissue but some of the results do, it is not a systematic search. I would recommend to exclude this or justify this - What would happen if this keyword had been included?

Experimental design

I am satisfied with the response to the comments.

Validity of the findings

I am satisfied with the response to the comments.

Additional comments

Two minor notes:
* Please be aware that I see no practical difference in distinguishing between RCC and clear cell RCC (line 231) or non-metastatic RCC or RCC. I would simplify everything to RCC.
* Disaccharides are typically named e.g. 0S CS - not 0s CS nor CS-0s. I would make sure this is consistently done throughout the manuscript.

Reviewer 2 ·

Basic reporting

No comment.

Experimental design

No comment.

Validity of the findings

No comment.

---

## Round 0.3 · accepted · Accept

Thanks for the extensive work on the manuscript. Please consider to optimize the disaccharide nomenclature throughout the manuscript

Reviewer 1 ·

Basic reporting

I am satisfied with the response to the comments.

Experimental design

No further comments.

Validity of the findings

No further comments.

Additional comments

Please note that my second comment was ignore as the CS and HS disaccharide names are not used consistently see line 11 pg 8 vs line 25 pg 7.